# Assessment of Oral Poliovirus Vaccine Viability and Titer at Delivery Points in Kinshasa, the Democratic Republic of the Congo: Implications for Cold Chain Management

**DOI:** 10.3390/vaccines13070680

**Published:** 2025-06-25

**Authors:** Gracia Kashitu-Mujinga, Anguy Makaka-Mutondo, Meris Matondo-Kuamfumu, Fabrice Mambu-Mbika, Junior Bulabula-Penge, Trésor Kabeya-Mampuela, Frida Nkawa, Grace Wanet-Tayele, Bibiche Nsunda-Makanzu, Pierre Nsele-Muntatu, Lusamba Kabamba, Antoine Nkuba-Ndaye, Aimé Mwana wa bene Cikomola, Elisabeth Mukamba-Musenga, Steve Ahuka-Mundeke

**Affiliations:** 1Département de Virologie, Institut National de Recherche Biomédicale, Kinshasa 01204, Democratic Republic of the Congo; anguymakaka90@gmail.com (A.M.-M.); merismatondo201@gmail.com (M.M.-K.); mambumbika2@gmail.com (F.M.-M.); juniorbulapenge@gmail.com (J.B.-P.); tresorkabens16@gmail.com (T.K.-M.); fridankawa@gmail.com (F.N.); gracewanet@gmail.com (G.W.-T.); bibinstel@gmail.com (B.N.-M.); antoinnkuba@gmail.com (A.N.-N.); amstev4@gmail.com (S.A.-M.); 2Service de Microbiologie, Département de Biologie Médicale, Cliniques Universitaires de Kinshasa, Université de Kinshasa, Kinshasa XI, Democratic Republic of the Congo; 3Graduate School of Biomedical Sciences, Institute of Tropical Medicine, Nagasaki University, Nagasaki 852-8523, Japan; 4Département de Biologie Médicale, Université Protestante au Congo, Kinshasa 01212, Democratic Republic of the Congo; 5Global Polio Eradication Initiative, Kinshasa 01204, Democratic Republic of the Congo; kabambal@who.int; 6Programme Elargi de Vaccination, Kinshasa 01204, Democratic Republic of the Congo; aimcik@yahoo.fr (A.M.w.b.C.); elisabethmukamba@gmail.com (E.M.-M.); 7Comite d’urgence Polio, Kinshasa 01204, Democratic Republic of the Congo

**Keywords:** vaccine viability, OPV, viral titer, cold chain, Kinshasa

## Abstract

Background: Poliomyelitis is a vaccine-preventable disease, with oral poliomyelitis vaccines (OPVs) and injectable poliomyelitis vaccines. In the Democratic Republic of the Congo (DRC), circulating vaccine-derived polioviruses (VDPVs) persist due to intrinsic and extrinsic factors, including the quality of the cold chain, which may make the vaccines less effective. This study’s objective was to evaluate the cold chain’s quality of OPVs and its effect on the vaccine’s viability and potency at different levels in health systems in Kinshasa. Methods: A cross-sectional study was conducted in Kinshasa, collecting OPVs at different levels of the health pyramid. Vaccine viability was assessed by cell culture using a modified World Health Organization (WHO) protocol, and the viral titer was determined using the Karber formula. The vaccine titer was classified as “very good”, “good”, or “poor” according to the WHO standard’s viral titer. Results: A total of 53 vaccines were collected and analyzed, compressing 38 bivalent oral poliomyelitis (bOPV) vaccines and 15 novel oral poliomyelitis vaccines, type 2 (nOPV2). The viral titer ranged from log10^5.8^ to log 10^7.3^ and from log10^5.4^ to log10^8.9^ for the nOPV2 and the bOPV, respectively. Of these 53 vaccine samples, 10% of the bOPVs showed viral titers below the recommended WHO threshold (>10^6^ CCID_50_/dose), 100% of the nOPV2 had viral titers within the WHO standards (>10^5^ CCID_50_/dose), and a significant decline in the viral titer was observed for both types of vaccines (nOPV2 and bOPV) as the distribution progressed along the level of the health pyramid. Conclusions: This study demonstrated that the viral titer of OPV declined from central to peripheral areas in routine and campaign strategies in Kinshasa.

## 1. Introduction

Poliomyelitis is an endemic–epidemic viral disease caused by the poliovirus. It is a highly contagious and debilitating disease characterized by acute flaccid paralysis, usually unilateral and affecting mainly the lower limbs, most commonly in children under 5 years of age. It can cause death in 10% of cases due to paralysis of the respiratory muscles. The wild poliovirus (WPV) serotypes WPV1, WPV2, and WPV3, which belong to the family *Picornaviridae* and the genus Enterovirus, have been identified as responsible for the epidemics [1,2,3].

Poliomyelitis remains a preventable disease by vaccination, thanks to the administration of oral poliovirus vaccines (OPVs) and injectable poliovirus vaccines (IPVs), which are included in the national immunization program of several countries, and the vaccine is highly immunogenic [4,5,6,7]. These vaccines have significantly reduced the number of poliomyelitis cases and led to the eradication of wild poliovirus serotype 2 (WPV2) by 2015 and serotype 3 (WPV3) by 2019 through the efforts of the global eradication program led by the World Health Organization (WHO) as part of the Global Polio Eradication Initiative (GPEI) [3,8,9,10,11,12,13].

The Democratic Republic of the Congo (DRC) was classified as an endemic zone in 1988, with 2.13 cases per million population per year, and was reclassified as a country certified free of wild poliovirus in 2015, after the WPV was eradicated in 2011. This achievement was made possible by the efforts of the Expanded Program on Immunization (EPI) in the Democratic Republic of the Congo (EPI-DRC), which was put in place in 1980. Since then, the immunization against poliomyelitis has remained one of the main EPI pillars in managing vaccine-preventable diseases in the DRC using routine and mass campaign immunization strategies [14,15,16,17].

Despite these significant advances in poliomyelitis eradication, the disease remains a global health problem due to persistent circulation of the wild poliovirus strain in some countries and reversion of the vaccine strain to vaccine-derived polioviruses (VDPVs) due to the mutations and genetic recombination of the attenuated poliovirus strains contained in OPVs with other enteroviruses, causing epidemics in regions free of wild poliovirus. These epidemics occur mainly in the context of inadequate immunization coverage in certain communities, particularly in Africa, including the DRC, which reported 263 cases of VDPV in 2023 [7,9,10,15,16,17].

In the DRC, efforts have been made to improve immunization coverage through routine immunization and mass immunization campaigns, which have resulted in over 85% coverage according to post-campaign surveys conducted by EPI-DRC [9,14,17]. Despite efforts to improve coverage, VDPV’s outbreaks, like other vaccine-preventable diseases, continue to occur in certain provinces in the DRC, including Kinshasa, where vaccination is carried out, leading to the resurgence of epidemics [10,12,16,18]. Several factors contribute to this resurgence of epidemics, including intrinsic factors related to the vaccinated individuals (such as immunodeficiency, malnutrition, and lack of vaccination compliance); extrinsic factors related to the pathogen (such as the emergence of strains not covered by the vaccine); and vaccine-related factors, in particular the quality and integrity of the cold chain [19,20,21,22,23,24].

The cold chain plays a crucial role in preserving the immunogenicity of the vaccine from production to the vaccination site. Proper storage of vaccines requires a strong and efficient logistics system, as well as qualified personnel, such as pharmacists, who play a central role in maintaining the cold chain, in particular temperature control, traceability of heat-sensitive products, and training of personnel involved in transporting and storing vaccines. Some vaccines, such as OPVs, require a cold chain capable of maintaining the required temperature range to ensure viability and maintain viral potency [25,26,27,28]. Nevertheless, effective maintenance of the cold chain remains a significant challenge in numerous countries, particularly those with low incomes, such as the DRC, where there are several challenges related to poor infrastructure and logistics, such as unreliable electricity, insufficient or poorly maintained cold chain equipment, and limited human resources and capacity.

However, it is essential to assess or monitor the cold chain for the storage of vaccines. This can be done either directly by closely monitoring the temperature, although this approach, despite technological advances, has several limitations, or indirectly by assessing the vaccine itself. The latter involves assessing the viability of the vaccine and the viral titer to determine whether there are still enough viral particles to trigger an immune response.

This study used the cell culture and viral titration approach, with the main objective of assessing the quality and integrity of the cold chain for vaccine storage and its impact on the viability and viral titer of OPV in Kinshasa, the capital of the DRC.

## 2. Materials and Methods

### 2.1. Study Type, Period, and Sites

This was a multicenter cross-sectional study conducted from January 2024 to June 2024, as part of an evaluation of vaccination activities. The study was conducted at vaccine storage points of the EPI-DRC. Thirty-seven sites were randomly selected in clusters at a central-level site, intermediate-level sites, and peripheral-level sites (see Figure 1).

### 2.2. Study Units

For the purposes of this study, the novel oral polio vaccine type 2 (nOPV2) and bivalent oral polio vaccine (bOPV) were selected for indirect evaluation of the cold chain, with the vaccine being stored at EPI vaccine storage sites. At least one representative vaccine from each available vaccine lot number at the site was used, with a minimum of two different lots numbers selected from each level of the storage health pyramid. Sampling was conducted based on both routine and mass campaign immunization strategies depending on availability of the vaccine and limited stock at the sites.

### 2.3. Data and Vaccines Sample Collection

The following variables were considered:

Vaccine-related data including the location of the vaccine sampling site, the type of vaccine, the manufacturer, the lot number, the expiration date, the receipt date of the vaccine at the vaccination site, the date of sample collection, the date of arrival at the laboratory, the vaccine vial monitor (VVM) stage, the vial condition (opened or closed), the type of storage equipment used for the vaccine, and the storage temperatures at the time of sample collection and upon arrival at the INRB laboratory,

Biological data including the cell culture results and the viral titer results.

Globally, data were collected using both pre-established data collection forms and RedCap software version 5.28.1. The collection of OPV vaccine samples from EPI sites was conducted in accordance with the WHO guidelines [28,29,30]. The vials were meticulously packaged in triple-layer packaging and transported to the laboratory in a cool box at −20 °C. The temperature was monitored using the LogTag^®^ recorders HAXO-8 version 1.0 device from the site to the INRB laboratory [31]. Upon arrival at the virology laboratory, the samples were stored at −20 °C under continuous monitoring until analysis.

### 2.4. Laboratory Analysis of Vaccine Samples

Prior to the execution of laboratory analyses, a preliminary phase was initiated with the objective of validating the protocol that would subsequently be employed for the testing of vaccine samples. Following the successful validation of the aforementioned protocol, the analyses were conducted using cell culture methods.

Validation of the Analysis Protocol

In this study, four ranges of analytical protocols were evaluated in accordance with the methodology outlined in the WHO Manual of Laboratory Methods for Vaccines Testing Used by the EPI (we named it WHO2), as well as that described in the Polio Laboratory Manual 4th Edition 2004: Immunization, Vaccines and Biologicals to assess the sensitivity of cell lines (we named it WHO1) [28,29].

Utilizing these established protocols, a control vaccine sample was examined. This sample was retrieved from the central storage facility and had been maintained under optimal conditions at a temperature of −20 °C. The sample was then subjected to testing across a range of vaccine dilution factors (see Table 1 and Table 2). This process was undertaken until a dilution factor that satisfied the validation criteria for a cell culture plate was identified. The validated protocols were designated as WHO1 for nOPV2 and WHO2 for bivalent OPV (bOPV).

Of all the protocols tested, the WHO protocol for the analysis of nOPV2 samples was adapted as follow: the first dilution of the vaccine was carried out at a concentration of 1/100 (instead of a first dilution of 1/10) the dilution factor was 100 (instead of 10); the dilution range of the vaccines was established between 10^−6^ and 10^−9^ (instead of 10^−4^ to 10^−7^) with a logarithmic difference of 1 between the dilution steps, in a total volume of vaccine-diluted mixture, in and L20B cells of 200 µL. This protocol was validated with a control sample (identified cDC, lot 2224523, Vaccine Vial Monitor Inner square is lighter than outer ring, expiration date not reached, taken at −20 °C from the central deposit) grown on the L20B cell (from CDC Atlanta, Lot 70029026 GR-924) at 2 × 10^5^ cells/mL cell concentration. The protocol was named WHO 1—This study.

The WHO protocol, which we adapted and named WHO2, was validated using the control sample (identified rDC152, lot 1803P152, Vaccine Vial Monitor Inner square is lighter than outer ring, expiration date not reached, taken at −20 °C from the central deposit) cultured on the L20B cell line (from CDC Atlanta, Lot 70029026 GR-924). The first validated dilution was 1/10, and the dilution factor was 10, within a vaccine dilution range of 10^−3.5^ to 10^−8^ with a logarithmic difference of 0.5 between the dilution steps, in a total volume of 200 µL vaccine diluent and cells (instead of a total volume of 300 µL vaccine diluent and cells). The modified protocol was used for the analysis of all remaining bOPV samples.

The following criteria were used to validate and interpret the cell culture results:

The plate was considered as valid if 50% to 100% of cytopathic effects (CPEs) were observed at the first considered dilution and 0% to 10% of CPEs at the last considered dilution, depending on nOPV2 or bOPV protocols. In contrast, the plate was considered as not valid when less than 50% of CPEs were observed at the first considered dilution and more than 10% of CPEs at the last considered dilution depending on nOPV2 or bOPV protocols.

After the plate was validated, the results were classified as “very good,” “good,” or “poor” based on the percentage of CPEs on the culture plate and the obtained viral titer. These were then compared with the manufacturer’s reference viral titers [nOPV2 (type 2): >10^5^ CCID_50_; bOPV (type 1): >10^6^ CCID_50_; bOPV (type 3): >10^5.8^ CCID_50_] (Table 3).

b.Oral Poliovirus live attenuated vaccine cell culture.

The OPV live attenuated vaccine was performed in five main steps, as follows:

Step 1: Preparation of permissive cells and working suspension (according to WHO protocol) [28]

Vaccine samples were cultured on L20B cells, a genetically modified mouse-derived cell line expressing the human poliovirus receptor, which allowed selective replication of the poliovirus. The cells were prepared in a growth medium composed of Minimal Essential Medium Eagle (HIMEDIA Ref.RNBL4184 Lot.0000984337) supplemented with 10% fetal bovine serum (gibco Ref.10091-14B, Lot.2577131P) to achieve a confluent layer of healthy, elongated, and compact cells. Cell passages below 15 times and 10 mL of the working cell suspension, containing 1 to 2 × 10^5^ cells/mL, were required per microtitration plate. This suspension was prepared in the maintenance medium using Minimal Essential Medium Eagle containing 2% fetal bovine serum, allowing the formation of a confluent monolayer cells within 2–3 days.

Step 2: Vaccine dilution in maintenance medium in accordance with the WHO protocol [28,29].

The maintenance medium (MM) was distributed into 1 to 10 different dilution tubes (15 mL Eppendorf tubes) according to the protocol, depending on whether the sample was a monovalent vaccine (nOPV2) or a bivalent vaccine (bOPV). The initial dilution 0.1 mL vaccine was mixed with a 9.9 mL diluent solution, and 0.2 mL vaccine was mixed with a 1.8 mL diluent solution for nOPV2 and bOPV, respectively; the diluted sample was then serially diluted through each subsequent dilution tube. The considered dilution ranges were 10^−6^ to 10^−9^ and 10^−3.5^ to 10^−8^ for nOPV2 and bOPV, respectively [Appendix A, Appendix B].

Step 3: Inoculation of diluted vaccines in to L20B cell line (from CDC Atlanta, Lot 70029026 GR-924)

In columns 1 to 10 of the sterile microtitration plate, 100 µL of the diluted nOPV2 vaccine of 10^−6^ to 10^−9^ dilution was inoculated into 100 µL of the L20B cell suspension plate [Appendix A]. Then, 50 µL of the diluted bOPV vaccine of 10^−3.5^ to 10^−8^ was inoculated into 100 µL of cell L20B suspension in a sterile, round-bottom, 96-well microtitration plate, which already contained 50 µL of maintenance medium [Appendix B]. In columns 11 and 12, 100 µL of L20B cells was mixed with 100 µL of maintenance medium and was considered to be a negative control. The plate was then hermetically sealed using adhesive tape (Scotch tape). Finally, the plate was incubated at 36 °C in 5–10% of CO_2_ atmosphere for 5 days.

Step 4: Monitoring cytopathic effects

The plates were examined on a daily basis from day 1 to day 5 using an inverted microscope to monitor the cytopathic effects (CPEs). The CPE manifestations included cell rounding, with cytoplasmic inclusions pushing the nucleus to the periphery, and cell destruction, leading to the detachment of the cell monolayer [28,29,30]. These effects are indicative of the viability of a live attenuated vaccine (see Figure 2).

Step 5: Calculation of viral titer [28,29]

On the 5th day of observation, using the lab bench sheet, we calculated the viral titer of the sample using Karber’s formula as follows:Log CCID_50_ = L − d (S − 0.5)
where L is the log of the lowest dilution used in the test, d is the difference between logarithmic dilution steps (0.5 for bOPV and 1 for nOPV2), and S is the sum of the proportion of positive tests (i.e., cultures showing CPEs).

The following parameters have been defined:

Cytopathic effects (CPEs) are morphological changes in cells caused by viral growth.

The viral titer (CCID_50_^®^) is the smallest amount of virus capable of causing cytopathic effects in 50% of infected cells.

An adequate viral titer is one in which the obtained viral titer is >10^5^ CCID_50_/0.1 mL for nOPV2 or >10^5.8^ CCID_50_/0.1 mL for bOPV.

An inadequate viral titer is when the obtained viral titer is ≤10^5^ CCID_50_/0.1 mL for nOPV2 or ≤10^5.8^ CCID_50_/0.1 mL for bOPV.

A viable vaccine is any vaccine that, after cell culture, exhibited 50–100% cytopathic effects at the first considered dilution range on the culture plate.

A very good vaccine is a viable vaccine with a 100% positive culture and a viral titer of >10^5^ CCID_50_/0.1 mL for nOPV2 or >10^5.8^ CCID_50_/0.1 mL for bOPV.

A good vaccine is a viable vaccine with a 50% positive culture and a viral titer of >10^5^ CCID_50_/0.1 mL for nOPV2 or >10^5.8^ CCID_50_/0.1 mL for bOPV.

A poor vaccine does not meet the criteria for a “good” or “very good” vaccine and is a viable vaccine with a 50% positive culture but a viral titer of ≤10^5^ CCID_50_/0.1 mL for nOPV2 or ≤10^5.8^ CCID_50_/0.1 mL for bOPV.

cDC is identified from a vaccine sample collected in the central deposit during the immunization campaign.

rDC 152 is an identified vaccine sample collected in the central deposit during the immunization routine.

### 2.5. Statistical Analysis

Data were collected using RedCap software, version 5.28.1, and exported to Microsoft Excel 2019, version 16.78. Data analysis was carried out using R software, version 4.4.1. Qualitative variables were presented as frequencies and proportions, while quantitative variables were presented as medians, interquartile ranges (IQRs), and the Kruskal–Wallis statistical method for carrying out the analysis of variance. The results are displayed in tables and figures.

### 2.6. Ethical Approval

This study was conducted as part of the evaluation of EPI related to poliomyelitis vaccination activities in the DRC and was carried out through a partnership between the INRB and the EPI-DRC. The ethical approval was obtained from the committee of Kinshasa University Public Health School (ethical approval number ESP/CE/62/2025).

## 3. Results

### 3.1. Protocol Results by Vaccine Type Against Validation Criteria

The WHO1-This study protocol was found to be 100% valid for nOPV2 vaccines and 100% invalid for bOPV. The evaluation of nOPV2 vaccines was facilitated by the aforementioned software. The WHO2 study protocol was found to be 100% valid for bOPV vaccines and 20% invalid for nOPV2. The evaluation of bOPV vaccines was conducted using this methodology (see Table 4).

### 3.2. Viability and Titer of OPV Vaccines According to the Level of the Health Pyramid

A total of 53 vaccine samples were collected and analyzed, including 15 nOPV2 and 38 bOPV during the study period. Of the nOPV2, 26.7% (4/15) were collected from vaccinators, while 28.9% (11/38) of the bOPV samples were collected from health zones, health centers, and vaccinators. The central level (A) exhibited a stable titer of 10^7.2^ CCID_50_/dose for nOPV2 and an average titer of 10^7.65^ CCID_50_/dose for bOPV; the intermediate level (B) demonstrated the highest average titer at 10^8.97^ CCID_50_/dose for bOPV; and the peripheral level displayed a progressive decrease in the mean titer for nOPV2 from 10^6.97^ CCID_50_/dose at the health zone level (Ca) to 10^6.35^ CCID_50_/dose at the vaccinator level (Cc), with smaller titer variations (Cc: 10^5.80^ to 10^6.80^ CCID_50_/dose). At the health center level (Cb), the bOPV exhibited the lowest average titer of 10^6.45^ CCID_50_/dose with a lowest minimum titer of 10^5.4^ CCID_50_/dose. In contrast, the Cc level had an average titer of 7.30, although it exhibited substantial titer variability from 10^5.70^ to 10^8.9^ CCID_50_/dose.

At central level A, 100% for both n OPV2 (1/1) and bOPV (2/2) vaccines was interpreted as very good. The same result was obtained at the intermediate level B for the bOPV vaccine. At health zone level Ca, 20% (1/5) of nOPV2 vaccines were classified as good, and 10% (1/11) of bOPV vaccines were classified as poor. At health center level Cb, 60% (3/5) of nOPV2 vaccines were classified as good, and 18% (2/11) of bOPV vaccines were poor. At the vaccinator’s level Cc, 100% (4/4) of nOPV2 vaccines were classified as good, while 10% (1/11) of bOPV vaccines were classified as poor (Table 5 and Figure 3 and Figure 4).

Level A had a single value indicating a lack of variability, with a vaccine titer of 7.2.

In level Ca, the median of viral titer was close to 7.0, and the interquartile range (IQR) was relatively wide, with a low outlier (~6.4).

In level Cb, the variability of viral titers was minimal, with a median of approximately 6.8, accompanied by a lower outlier of around 6.4.

Level Cc had a wide range of vaccine titers (~6.0 to ~7.2), with a median that approached 6.4.

H statistic = 6.40 and *p*-value = 0.094 (>0.05): no statistically significant difference was found at *p* < 0.05, although a trend was noted.

In level A, the titers were concentrated between approximately 7.5 and 8, with minimal variability.

Level B demonstrated the highest viral titers, with a median of around 9 and greater variability.

Level C exhibited a wider spread, with a median viral titer of 7.8 but values ranging from 6 to 8.

Level D had a median close to 6.5 and a few outliers.

Level E had a median around 7.5 but with greater variability.

H statistic = 13.7 and *p*-value = 0.008 (≤0.05): there was a statistically significant difference in bOPV viral titers between the different levels of the health pyramid.

### 3.3. Viability and Titer of OPV Vaccines According to Immunization Strategies

The viral titers of the nOPV2 vaccines utilized during the vaccination campaign ranged from 10^5.8^ to 10^7.3^ CCID_50_/dose. Based on culture and viral titer, 60% (9/15) of the nOPV2 vaccines were classified as good, and none were deemed to be poor.

However, the viral titers of the bOPV vaccines utilized during routine immunization ranged from 10^5.4^ to 10^8.9^ CCID_50_/dose. A total of 10.5% of bOPV vaccines (4/38) were categorized as poor, while 44.7% of bOPV vaccines (17/38) were classified as good or very good, as illustrated in Table 6.

### 3.4. Summary of Evaluations of the Two Immunization Strategies According to the Level of the Health Pyramid with OPV Vaccines and Study Sites

For the campaign strategy (nOPV2 vaccine), at the central level, 100% was very good; at the health zone level, 80% was very good, and 20% good; at the health center level, 40% very good, and 60% was good; and 100% of vaccinators’ nOPV2 was good. For the routine strategy (bOPV vaccine), 100% was very good at the central and intermediate levels; at the health zone level, 10% was bad; at the health center level, 18% were classified as poor, i.e., 2/11; 10% of vaccinators’ bOPV were poor. Furthermore, 8.1% (3/37: Binza Météo health zone, Kasavubu Health Centre and Kasavubu’s vaccinator, Kinsenso Gare Health Centre) of sites had inadequate vaccines (with titers below the WHO reference threshold) (Figure 5).

## 4. Discussion

This study provides evidence of the quality of the OPV vaccine storage cold chain at different levels of the heath pyramid, particularly in the field, during both campaigns and routine immunization activities in Kinshasa, the capital city of the DRC.

The study collected a total of 53 vials, including 15 vials of nOPV2 obtained during the immunization mass campaign and 38 vials of bOPV collected during routine immunization activities. Logistical constraints, including vaccine stockouts at specific sites and the inability to reach all study sites simultaneously during the immunization, particularly for vaccinators, justified the difference in the size of both nOPV2 and bOPV. However, the sample size used in the present study was sufficient for titer and viability evaluation according to the literature [25,26,27].

At the central level, the viral titer was 10^7.2^ CCID_50_/dose for n OPV2, while the mean titer of bOPV was 10^7.65^ CCD_50_/dose for b OPV, indicating that the vaccine was stable, and the initial storage condition (−20 °C) was satisfactory [28,29,30].

This outcome was consistent with those reported by Zipursky et al. in 2011 in Chad [25], who examined the stability of two control vaccines stored under identical conditions. The intermediate level exhibited the highest mean titer of 10^8.97^ CCID_50_/dose for bOPV, attributable to the more dependable cold chain (a negative cold room at –20 °C) and the stringent temperature control implemented at this level. However, the present study observed a progressive decline in viral titer as the central level was transitioned to the peripheral level, for both nOPV2 and bOPV utilized during the mass campaign and routine immunization, respectively. The p-value of viral titer for nOPV2 was close to the threshold (0.05), suggesting a trend toward a difference. With a larger sample size, a significant difference might emerge. The graphical analysis (Figure 3) showed an apparent decrease in titers from level A to level Cc.

At the peripheral level, this study demonstrated a decline in the mean viral titer for nOPV2 range from 10^6.97^ CCID_50_/dose at the health zone level to 10^6.35^ CCID_50_/dose at the vaccinator level (Cc), with smaller titer variations (10^5.80^ to 10^6.80^ CCID_50_/dose), which suggested a gradual loss of vaccine stability, which might be related to less stringent storage conditions, particularly temperatures above +8 °C. The mean viral titer found was consistent with the mean titer of 10^6.25^CCID_50_/dose found by Muhammad et al. in Nigeria during exposure to +60 °C in vaccine samples from the local government [26] and the mean titer of 10^6.2^ CCID_50_/dose found by Zipursky et al. in Chad after the vaccine had been exposed to ambient temperature for 2 days [25]. For bOPV, at the health center level, we found vaccines with the lowest mean titer of 10^6.45^ CCID_50_/dose and the lowest minimum titer of 10^5.4^ CCID_50_/dose, while at the vaccinator level we found high titer variability ranging from 10^5.70^ to 10^8.90^ CCID_50_/dose, suggesting a failure of the cold chain at the peripheral level, where storage conditions were uneven (ranging from refrigerators at the health zone and health center levels to vaccine carriers sometimes without ice packs at the vaccinator level). These findings are of major public health relevance as they highlight critical weaknesses in the last mile of the vaccine supply chain that could compromise immunization efforts. Ensuring potent vaccines reach the population, especially in remote or underserved areas, is crucial for preventing poliovirus transmission and achieving eradication goals in the DRC. They also imply that failure in maintaining recommended cold chain temperatures may contribute to the re-emergence of vaccine-derived polioviruses (VDPVs), thus reinforcing the urgency to strengthen cold chain infrastructure and monitoring. This is consistent with findings from Muhammad et al. in 2010, Zipursky et al. in 2011, and Eswaran et al. in 2003, who found a titer of 10^6.90^ CCID_50_/dose at +60 °C [26], a titer of 10^5.6^ CCID_50_/dose below the threshold of 10^5.80^ CCID_50_/dose after exposure to ambient temperatures [25], and vaccines at the threshold of 10^5.80^ CID_50_/dose [27], respectively.

In this study, the campaign immunization strategy demonstrated that vaccines met the expectations of the vaccination program, with viral titers exceeding the WHO reference threshold of 10^5^ CCID_50_/dose [28,29]. These titers exhibited relative homogeneity, with a marginal decline at lower levels. All nOPV2 vaccines were deemed adequate (ranging from very good to good). This outcome was consistent with those reported by Zipursky et al. in Chad, who found similar viral titers of 10^5.60^ to 10^6.40^ CCID_50_/dose [25]. The decline in viral titer at the peripheral level of the health pyramid might be related to low vigilance in temperature monitoring by vaccinators. These results further demonstrate that mass vaccination campaigns using nOPV2 can achieve better control of cold chain quality compared with routine services, which is an important insight for planning national immunization strategies.

In the context of the routine strategy, 10% of vaccines had titers below the WHO reference threshold of 10^5.80^ CCID_50_/dose [27], which was already observed at the peripheral level (health zones and vaccinator), with the lowest titer recorded at 10^5.4^ CCID_50_/dose, categorizing this as inadequate (poor vaccines). This outcome aligned with the findings reported by Zipursky et al. in Chad, where vaccines were exposed to ambient temperatures (10^5.60^ CCID_50_/dose), and by Muhammad et al. in 2010, where vaccines were exposed to +60 °C in the laboratory in Nigeria [25,26]. However, this result diverges with finding reported by Eswaran et al. in India, who found a range of 10^5.83^ to 10^6.49^ CCID_50_/dose [27]. All these findings point out the exposure of vaccines to temperatures out of the WHO range [28,29] at the sites due to poor storage conditions and a lack of temperature monitoring in the storage equipment. Continuous monitoring of the quality of the cold chain is essential to ensure that vaccines remain effective until they are administered. This can be done by performing regular temperature checks and assessing the viability of the vaccines.

Our results indicated a sequential deterioration of the cold chain across three distinct levels of vaccine supply and storage in Kinshasa. This indicated that the issue could have occurred from the health zone that supplied the health center, which in turn supplied the vaccinator within the same geographical area. These findings support the need to prioritize investment in cold chain infrastructure and monitoring in low-performing areas to ensure the equity and effectiveness of immunization programs across the DRC.

In the present study, the individual serotype was not subjected to titration. Had this been conducted, it would have yielded insights into the potency of the individual vaccine strain for bOPV, i.e., type 1 and type 3. However, given that the total titer was set in OPV, estimating the composite titer was sufficient to evaluate the b OPV. It should be noted that in this study, the sample was limited to the capital region, and only a single manufacturer of vaccine was used.

This study has the merit of being the first pilot study to assess the viability of a vaccine in the DRC in the context of the re-emergence of poliomyelitis caused by the vaccine strain, despite significant vaccination efforts. It also underscores the importance of cold chain management in vaccine storage to ensure the effectiveness of the immunization program in the DRC. Its findings are critical for guiding national strategies to close operational gaps and ultimately interrupt the transmission of vaccine-derived polioviruses in the country.

This study suggests to conduct similar study in other provinces where poliomyelitis is also occurring.

## 5. Conclusions

The study revealed a significant decline in vaccine viability from the central level to the peripheral level, particularly in routine strategy. While these findings cannot be generalized, they contribute to the understanding of the rise in vaccine-preventable diseases despite high vaccine coverage. The findings are of particular significance for decision-makers as they underscore the necessity for improvements in vaccine storage, especially in low-income countries. The results underscore the need for further studies in DRC provinces where poliomyelitis outbreaks are still recurrent.

## Figures and Tables

**Figure 1 vaccines-13-00680-f001:**
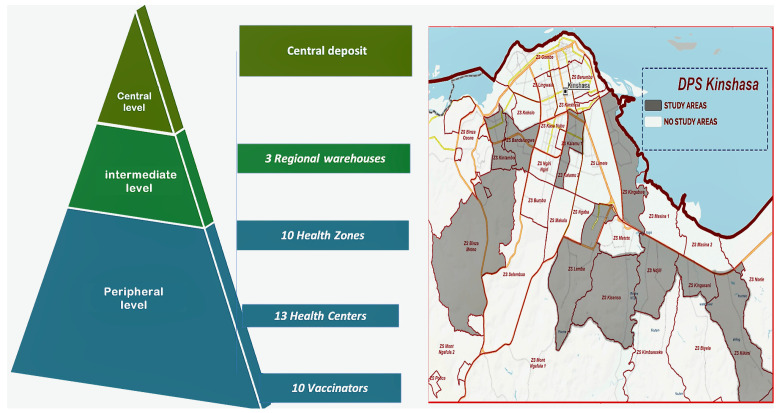
Mapping of the health pyramid and study sites (DPS: Division Provinciale de la Sante).

**Figure 2 vaccines-13-00680-f002:**
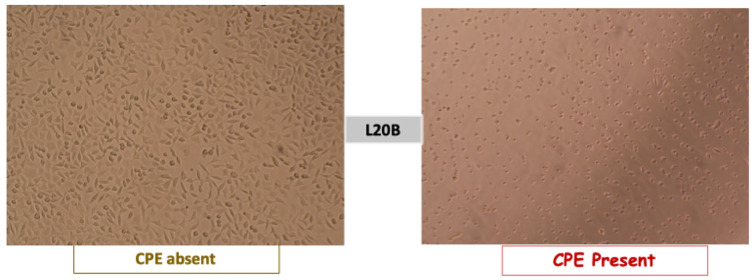
Poliovirus cytopathic effect observed under an inverted microscope in the L20B cell line. Source: study photo library.

**Figure 3 vaccines-13-00680-f003:**
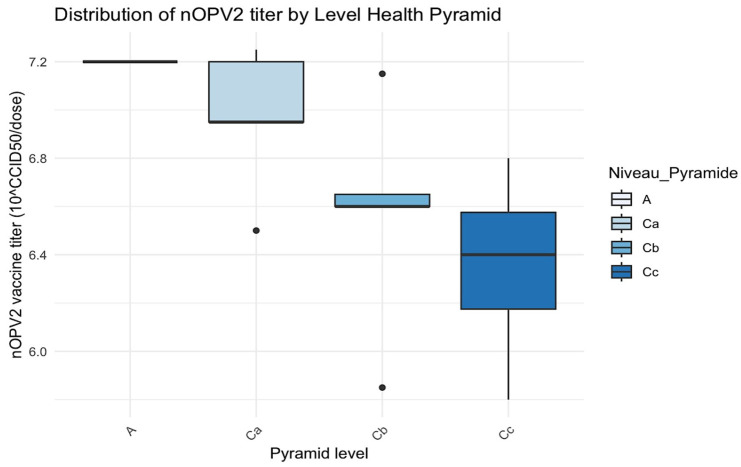
Distribution of nOPV2 titer by level of the health pyramid. The black dots are the outliers.

**Figure 4 vaccines-13-00680-f004:**
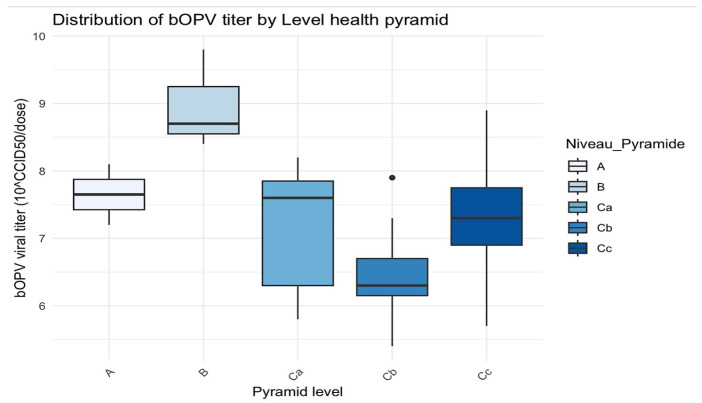
Distribution of bOPV titer by level of the health pyramid. The black dots are the outliers.

**Figure 5 vaccines-13-00680-f005:**
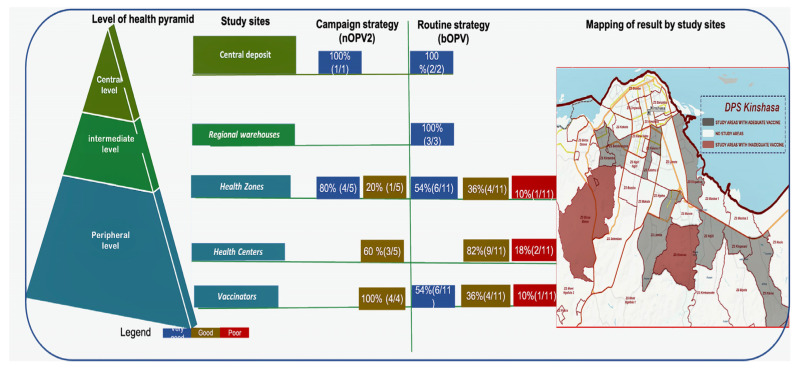
Schematic summary of OPV vaccine results by immunization strategies and geographical mapping of vaccines.

**Table 1 vaccines-13-00680-t001:** Validation of Protocols for nOPV2.

ID Vaccine	Vaccine Titer (LogCCID_50_)	Valid Criteria	Cell Culture				Protocol
			% CPE (First–Last Dilution)	Vaccine Dilution	Volume Per Well (uL)	Interval Dilution (1–10)	
cDC	8.55	No	100–30	1/10	200	6–9	WHO 1
cDC	7.8	Yes	100–0	1/100	200	6–9	WHO1—This study
cDC	7.2	Yes	90–0	1/100	300	6–9	Muhammad—This study
cDC	6.4	Yes	100–0	1/10	200	3.5–8	WHO 2

**Table 2 vaccines-13-00680-t002:** Validation of protocols for bOPV (type 1 and type 3).

ID Vaccine	Vaccine Titer (LogCCID_50_)	Valid Criteria	Cell Culture				Protocol
			% CPE (First–Last Dilution)	First Vaccine Dilution	Volume Per Well (uL)	Interval Dilution (1–10)	
rDC 152	9.5	No	100–100	1/10	200	6–9	WHO1
rDC 152	9.15	No	100–65	1/100	200	6–9	WHO1—This study
rDC 152	8.1	Yes	100–10	1/100	300	6–9	Muhammad—This study
rDC 152	7.2	Yes	100–0	1/10	200	3.5–8	WHO2

**Table 3 vaccines-13-00680-t003:** Interpretation of results.

% Cell Culture Per Plate		Viral Titer vs. Reference Viral Titer ^a,b^	Interpretation
First Dilution	Last Dilution		
80–100	0–10	Viral titer > reference viral titer	Very good
50–10	0–10	Viral titer > reference viral titer	Good
50–10	0–10	Viral titer ≤ reference viral titer	Poor

Legend: ^a^: Reference viral titer for nOPV2 (type 2: >10^5^ CCID_50_/dose); ^b^: Reference viral titer for bOPV (type 1: >10^6^ CCID_50_/dose, type 3: >10^5.8^ CCID_50_/dose).

**Table 4 vaccines-13-00680-t004:** Result of protocol validation.

Type Vaccine	Protocol	Criteria	
		Sample Valid n/N (%)	Sample Not Valid n/N (%)
nOPV2	WHO2	8/10 (80)	2/10 (20)
	WHO1—This study	10/10 (100)	0/10 (0)
bOPV	WHO2	10/10 (10)	0/10 (0)
	WHO1—This study	0/10 (0)	10/10 (100)

Legend: nOPV2—new oral polio vaccine type 2; bOPV—bivalent oral polio vaccine.

**Table 5 vaccines-13-00680-t005:** Viability and viral titer of OPV vaccines at different levels of the health pyramid.

Health Pyramid Level	Type of Vaccine n/N(%)		Viral Titer Average (10^xx^ CCID_50_/Dose) ^a,b^ (min–max)		Interpretation in Laboratory (%)					
	nOPV2 15/53 (28.3)	bOPV 38/53 (71.7)	nOPV2	bOPV	nOPV2			bOPV		
					Very Good	Good	Poor	Very Good	Good	Poor
A	1/15 (6.7)	2 (5.3)	7.2 (single value)	7.65 (7.20–8.10)	100	0	0	100	0	0
B	-	3 (8.0)	-	8.97 (8.40–9.80)	-	-	-	100	0	0
Ca	5 (33.3)	11 (28.9)	6.97 (6.50–7.25)	7.21 (5.80–8.20)	80	20	0	54	36	10
Cb	5 (33.3)	11 (28.9)	6.57 (5.85–7.15)	6.45 (5.40–7.90)	40	60	0	0	82	18
Cc	4 (26.7)	11 (28.9)	6.35 (5.80–6.80)	7.30 (5.70–8.90)	0	100	0	54	36	10

Legend: A—central level (central deposit); B—intermediate level (branches); C—peripheral level (Ca—health zones; Cb—health centers; Cc—vaccinators); nOPV2—new oral polio vaccine type 2; bOPV—bivalent oral polio vaccine; (10^xx^ CCID_50_/Dose) ^a^: viral titer > 10^5^ CCID_50_/dose: the titer listed by the manufacturer on the nOPV2 vaccine vial; (10^xx^ CCID_50_/Dose) ^b^—viral titer > 10^5.8^ CCID_50_/dose and viral titer > 10^6^ CCID_50_/dose: the titer listed by the manufacturer on the bOPV vaccine vial (type 1 and type 3).

**Table 6 vaccines-13-00680-t006:** Viability and titer of OPV vaccines by immunization strategies.

Immunization Strategy	% nOPV2 Culture (First–Last Dilution)		Viral Titer nOPV2 ^a^ (10^xx^ CCID_50_/Dose)	Total Samples (N = 15)	Interpretation
Campaign	100–10		6.9 ± 0.3	6/15 (40%)	Very good
	80–100	0–10			
	50–10		6.4 ± 0.4	9/15 (60%)	Good
	50–79	0–10			
	50–10		≤5	0	Poor
	50–79	0–10			
**Immunization** **Strategy**	**% bOPV Culture** **(First–Last Dilution)**		**Viral Titer bOPV ^b^** **(10 ^xx^ CCID_50_/Dose)**	**Total Samples (N = 38)**	**Interpretation**
Routine	100–10		8.2 ± 0.7	17/38 (44.74%)	Very good
	80–100	0–10			
	50–10		6.7 ± 0.6	17/38 (44.74%)	Good
	50–79	0–10			
	50–10		5.7 ± 0.2	4/38 (10.5%)	Poor
	50–79	0–10			

Legend: nOPV2—new oral polio vaccine type 2; bOPV—bivalent oral polio vaccine; (10^xx^ CCID_50_/Dose) ^a^—viral titer > 10^5^ CCID_50_/dose: the titer listed by the manufacturer on the nOPV2 vaccine vial; (10^xx^ CCID_50_/Dose) ^b^—viral titer > 10^5.8^ CCID_50_/dose and viral titer > 10^6^ CCID_50_/dose: the titer listed by the manufacturer on the bOPV vaccine vial (type 1 and type 3).

## Data Availability

Lab analysis and culture data are available on request from the corresponding and senior authors.

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
