# Peer review of "Assessment of Oral Poliovirus Vaccine Viability and Titer at Delivery Points in Kinshasa, the Democratic Republic of the Congo: Implications for Cold Chain Management"

_vaccines, 2025, doi:10.3390/vaccines13070680_

Round 1
Reviewer 1 Report
Comments and Suggestions for Authors
Kashitu-Mujinga and colleagues report a study of the stability of live oral poliovirus vaccine (the standard bivalent preparation with serotypes 1 and 3, and the nOPV type 2) used in routine immunization and immunization campaigns in the Democratic Republic of the Congo. They collected the samples from 111 storage points along the vaccine distribution chain from the central storage facility down to the regional field offices, and assessed the virus titer using the TCID50 titration method. The results demonstrate a general well preservation of the vaccine viability, however, 10% of the samples of the standard OPV collected at the end point distribution facilities revealed a titer below the WHO recommended threshold, which may result in suboptimal immunization.
The study is well-performed and provides important information on the challenges of poliovirus vaccine delivery and distribution in resource-limited settings. The detailed Methods section deserves particular praise, but the manuscript needs to be edited for clarity:
Methods. Check Table 1 headings (Volume, not Volum), other headings are apparently shifted.
Results, Table 1. Please explain the significance of the “invalid protocol” results for bOPV. Does it mean this protocol was not used in subsequent vaccine viability evaluations? If you continued with this protocol, what was the point of this “validation”?
Table 2. What do the numbers in the “XX Culture(%)” columns represent? How are they related to the summarized data in the last two columns?
Line 58. should be immunogenic, not immunogen
Lines 148-149. This sentence seems to be clipped
Line 285, remove “respectively”. The data presented (11/38 of the samples) do not allow attribution to individual collection places.
Line 342. Remove “no nOPV2 vaccine was bad”, this is a restatement of the previous description of the data.
Discussion needs to be streamlined, it reiterates the same statements over and over. Sections between lines 392-396 and 410-412 are almost direct repeats.
Author Response
please see the attachement

Reviewer 2 Report
Comments and Suggestions for Authors
Poliomyelitis is vaccine-preventable, but circulating vaccine-derived polioviruses persist in DRC due to cold chain issues. This Kinshasa cross-sectional study analyzed 53 OPVs (38 bOPV, 15 nOPV2). Using WHO protocols, 10% had titers below limits; nOPV2 showed higher titers than bOPV, but titers declined at lower health levels, especially in remote areas, indicating cold chain vulnerabilities in distribution.
Here are some of my suggestions and questions:
1, Lines 39-40. Part of the data is expressed vaguely such as "10% shown viral titers below WHO recommended limits". What are the WHO recommended limits? And clarify the proportion of unqualified vaccines for different types, such as how much nOPV2 is unqualified and how much bOPV is unqualified?
2, Lines 43-44. The conclusion section mentions "viability and viral titers of OPV vaccines decline from central to peripheral areas", but does not explicitly mention the differences between different vaccine types (nOPV2 and bOPV).
3, Lines 264-268. The statistical analysis section (Section 2.5) only mentions the use of R software, but does not specify the specific statistical methods (such as whether to conduct variance analysis, correlation analysis, etc.)
4, The descriptions of Figure 3 and Figure 4 are relatively brief and do not specify the vertical coordinate unit (such as CCID50/dose). Suggest adding vertical coordinate units and analyzing whether there are significant differences in titer changes between different levels.
5, Suggest a deeper discussion on the significance of the research results, such as their implications for polio prevention and control in the DRC.
6, It is suggested to clarify the limitations of this study in the discussion, such as "the sample is limited to the capital region and only a single manufacturer of vaccine is used, and extrapolation of results should be cautious.
Reviewer 3 Report
Comments and Suggestions for Authors
1) The Authors performed an article entitled "Assessment of Oral Poliovirus Vaccine Viability and Titer at Delivery Points in Kinshasa, the Democratic Republic of the Congo: Implications for Cold Chain Management". The main aim of the study was to evaluate the cold chain’s quality of OPVs and its effect on the vaccine’s viability and potency at different levels on health system in Kinshasa. In addition, the School of Public Health’s Ethics Committee at the University of Kinshasa provided consent for the data to be used in scientific publications following the ethical approval number ESP/CE/62/2025.
2) All abbreviations used must be revised. For example, please see "OPV" in lines 56 and 92.
3) Please clearly highlight the major innovation/strong point of the article.
4) Please mention the main limitations of the study without diminishing its value.
5) Lines 90-98: "cold chain" - please highlight the role of the pharmacist as the ultimate expert on the medicinal product. The physical, chemical, microbiological and biopharmaceutical stability of vaccines, during distribution and storage, is only fully ensured if the pharmacist is fully involved in the circuit. Please justify your answer.
6) Line 132: not complete...
7) Line 158: Please indicate the Table number in the caption...
8) Line 169: Table 2?
9) Line 189: Table 3?
10) Line 225: Finally, the plate was incubated at 36°C in 5-10 % of COâ‚‚ atmosphere for 5 days. Why did you use these conditions? Please justify your answer.
11) Line 350: Figure 1 or Figure 5?
12) Lines 427-429: How many vaccine manufacturers are there? It is necessary to characterize the sector...
Round 2
Reviewer 2 Report
Comments and Suggestions for Authors
The author responded well to the reviewer's comments, and I have no further suggestions or comments.